# Nano-Theranostics Constructed from Terpyridine-Modified Pillar [5]arene-Based Supramolecular Amphiphile and Its Application in Both Cell Imaging and Cancer Therapy

**DOI:** 10.3390/molecules27196428

**Published:** 2022-09-29

**Authors:** Youjun Zhou, Lu Yang, Longtao Ma, Ying Han, Chao-Guo Yan, Yong Yao

**Affiliations:** 1School of Chemistry and Chemical Engineering, Yangzhou University, Yangzhou 225001, China; 2School of Chemistry and Chemical Engineering, Nantong University, Nantong 226019, China

**Keywords:** pillar [5]arene, host–guest chemistry, terpyridine, cancer therapy, self-assembly

## Abstract

Theranostics play an important role in cancer treatment due to its realized real-time tracking of therapeutic efficacy in situ. In this work, we have designed and synthesized a terpyridine-modified pillar [5]arenes (**TP5**). By the coordination of terpyridine and Zn^2+^, the complex **TP5**/Zn was obtained. Then, supramolecular amphiphile can be constructed by using host–guest complexation between a polyethylene glycol contained guest (**PM**) and **TP5/Zn**. Combining the fluorescence properties from the terpyridine group and the amphiphilicity from the system, the obtained **TP5/Zn/PM** can further be self-assembled into fluorescent particles with diameters of about 150 nm in water. The obtained particles can effectively load anti-cancer drugs and realize living cell imaging and a precise release of the drugs.

## 1. Introduction

Pillar[n]arenes are one type of classical macrocycles, which are synthesized from the hydroquinone-derivatives connected at 2,5-position by methylene groups [1,2,3,4]. Pillar[n]arenes have a rigid hydrophobic cavity with an adjustable size, which endows the pillar[n]arenes with rich host–guest properties [5,6,7,8]. In recent years, the synthesis, functionalized derivation, host–guest properties and applications of pillar[n]arenes have attracted much attention [9,10,11,12,13,14,15,16,17,18], especially in the aspect of constructing pillar[n]arene-based controllable drug release systems for cancer therapy [19,20,21,22,23,24,25]. For example, Prof. Pei constructed tumor microenvironment responsive supramolecular glyco-nanovesicles based on a diselenium-bridged pillar [5]arene dimer for targeted chemotherapy [19]. Prof. Fan fabricated multifunctional supramolecular vesicles for combined photothermal/photodynamic/hypoxia-activated chemotherapy [24]. Prof. Wang and coworkers prepared a dual acid-responsive bola-type supramolecular vesicle by the complexation between a water-soluble pillar [5]arene (WP5) and an acid-sensitive guest molecule (G) containing the 2,4,8,10-tetraoxaspiro [5.5]undecane moiety for an efficient intracellular anticancer drug delivery [26]. Our group prepared the cationic water-soluble pillar [5]arene-modified Cu_2–x_Se nanoparticles for targeting photothermal therapy in the NIR-II window [25].

Terpyridine not only has a strong fluorescence signal under suitable excitation conditions, but also is used as a general ligand to form transition metal complexes; this provides a new idea for the preparation of living cell fluorescent imaging biomaterials [27,28]. After the coordination of ligands with metal ions, the binding mode and ability of the metal complexes with the target are affected by the structural effect of the ligands and the electronic effect of the metal ions, and as the introduction of metal ions can produce a synergistic effect, the activity of the complex is increased [29]. On the other hand, some transition metal ions are very important to the human body, for example, zinc plays an indispensable role in many life processes: it can promote the metabolism of various substances in the body, promote the proliferation of the lymphatic system, enhance the resistance to viruses and bacteria, etc. [30,31].

Herein, we have designed and synthesized a terpyridine-modified-pillar [5]arene (**TP5**). By the coordination of terpyridine and Zn^2+^, pillar [5]arene-Zn complexes (**TP5/Zn**) were obtained. Then, by using a host–guest complexation between **TP5/Zn** and a guest molecule which contains polyethylene glycol (**PM**), the supramolecular amphiphile (**TP5/Zn/PM**) was constructed successfully. **TP5/Zn/PM** can further self-assemble into fluorescent particles with a diameter of about 150 nm due to its combined amphiphilic nature and fluorescence properties from the terpyridine group. The obtained fluorescent particles can load and control the release of anti-cancer drugs effectively to realize the precise release of drugs and living cell imaging. This work may provide a new way for scientists to construct nano-theranostics through dynamic host–guest interactions.

## 2. Results and Discussion

### 2.1. Characterization of TP5

As described in Figure 1, **TP5** was prepared from a mono alkyl bromide-modified pillar [5]arene (BrP5) and terpyridine in CH_3_CN with KI as the catalyst. The structure of **TP5** was fully characterized by conducting ^1^H NMR (see Appendix A), ^13^C-NMR (see Appendix A), MS (see Appendix A) and single crystal X-ray analyses. No proton signal was observed below 0 ppm from the ^1^H NMR spectra of **TP5s**, indicating that the alkyl chain is outside the cavity of pillar [5]arene. The single crystal structure of **TP5** clearly shows that the alkyl chain is not penetrated into the cavity, which is consistent with the NMR results (Figure 1).

### 2.2. Coordination and Host–Guest Recognition

As we all know, terpyridine is a classical tridentate ligand, which can coordinate with a variety of metal ions to form metal complexes [27]. On the other hand, Zn^2+^ ions play an indispensable role in many life processes. In order to explore the recognition performance of **TP5** on Zn^2+^ ions, the fluorescence of **TP5** was investigated after adding Zn^2+^ ions. As shown in Appendix A, compared with **TP5**, the fluorescence intensity decreased sharply after the addition of ZnCl_2_. Then the fluorescence titration of Zn(II) with **TP5** showed a continuous decrease with the increase in the Zn(II):**TP5** ratio (from 0:10 to 10:0 µM). The fluorescence titration curve revealed that the fluorescence intensity at 406 nm decreased linearly on increasing the ratio of the Zn(II) ions (Figure 2a). The method of continuous variation (Job’s plot) was also performed to prove the 1:1 stoichiometry (Figure 2b). All the above results indicated that **TP5** and Zn^2+^ could form a stable 1:1 complex (**TP5**/Zn).

The free cavity in **TP5/Zn** endowed it with a unique host–guest property. **PM1** was chosen as the model guest to investigate the host–guest interaction between **TP5/Zn** and the guest molecule which contains polyethylene glycol. The proton NMR spectrum of an equimolar solution of **TP5/Zn** and guest **PM1** showed that the complex is in fast exchange on the proton NMR time scale (see Appendix A). Protons Ha, Hb, Hc and Hd on guest **PM1** shifted upfield after complexation while no obvious chemical shift changes were observed for He after complexation. These phenomena suggested that **PM1** was threaded through the cavity of **TP5/Zn** to form a [2]pseudorotaxane with the imidazolium part located in the cavity and the tail (He) out of the cavity. Furthermore, the Job’s plot analysis revealed that a stoichiometry system between **TP5/Zn** and **PM1** in the complex can be determined at 330 nm (see Appendix A). As expected, when the molar fraction of the complex sensor was 0.5, the absorbance reached a maximum, which demonstrates that the interaction between **TP5/Zn** and **PM1** forms a 1:1 complex.

### 2.3. Construction and Characterization of Nano-Theranostics

After the establishment of the new host–guest recognition motif in aqueous solution, we further applied it to construct the supramolecular amphiphile **TP5/Zn/PM**. The guest molecule **PM** has two advantages, the first is that it has a strong complexing ability with pillar [5]arene, and the second is that it endows the system with amphiphilicity, which makes the system form stable assemblies. The critical aggregate concentration (CAC) in water was determined to be about 2.32 × 10^−6^ mol/L from the change of water surface tension (Figure 3a). Furthermore, the dynamic light scattering (DLS) experiment performed with a 2.50 × 10^−6^ M aqueous solution of **TP5/Zn/PM** over a scattering angle of 90°, showed a narrow size distribution (Figure 3b). The average hydrodynamic diameter (Dh) of **TP5/Zn/PM** was observed to be about 150 nm. Transmission electron microscopy (TEM) was used to investigate the morphology of **TP5/Zn/PM**-based assemblies in water. As shown in Figure 3c, when **TP5/Zn/PM** was dissolved in water, it self-assembled into spherical structures with a diameter of about 140 nm immediately. In addition, the SEM image showed that **TP5/Zn/PM** self-assembled into particles in water, which consisted with the TEM results (Figure 3d). **TP5/Zn/PM** self-assembled into particles in water possibly due to a larger curvature of its membrane, and a membrane with a larger curvature could form micelles easily. Furthermore, the zeta potential of hollow vesicles was −42.3 ± 4.78 mV, indicating that the vesicles were very stable in solution [32].

### 2.4. Cell Imaging

Then, the Dox loading and in vitro release were carried out to check whether **TP5/Zn/PM**-based particles can be used as a drug carrier. The particles entrapped Dox effectively with an encapsulation efficiency of 195 μg/mg (Appendix A), indicating that **TP5/Zn/PM**-based particles are satisfactory drug-loaded materials. The Dox release experiments were investigated in PBS with pH 7.4, 6.0 and 4.7, respectively (Appendix A). Then, the UV-Vis spectra was used to monitor the release of Dox against time. After 10 h, the total release rate was 6.9% at pH 7.4, 45.9% at pH 6.0 and 62.1% at pH 4.7, respectively. As we all know, the microenvironment of tumor tissue is acidic due to an excess in expressed lactic acid and CO_2_ in the metabolites of tumor cells [33,34]. Dox loading particles can suspend the Dox release in normal cells, and it is found that this pH-responsive Dox release is a slow process under acidic conditions. Therefore, Dox loading particles can prolong dosing time and reduce toxicity.

Cellular uptake ability is an important parameter for the therapeutic effects of nanomaterials. [35,36,37] On the other hand, the terpyridine unit in pillar [5]arene has a strong fluorescence signal under a suitable excitation, so we can utilize a laser scanning confocal microscope (CLSM) to investigate the internalization of the obtained materials by HeLa cells. The HeLa cells were incubated in Dulbecco’s modified Eagle’s medium (DMEM). The medium was supplemented with 10% fetal bovine serum and 1% penicillin-streptomycin. The HeLa cells were seeded in 96-well plates (5 × 10^4^ cell mL^−1^, 0.1 mL per well) for 24 h at 37 °C in 5% CO_2_. Then the cells were incubated in **TP5/Zn/PM**, Dox and **TP5/Zn/PM**/Dox for 4 h, respectively. The medium was then removed, and the cells were washed 3 times with a phosphate buffer. Finally, the cells were observed by fluorescence microscopy. As shown in Figure 4, the cells that were treated with **TP5/Zn/PM** exhibited a bright blue fluorescence emission in the cytoplasm, while those treated with Dox exhibited a bright red fluorescence emission in the nucleus. However, the cells that were incubated with **TP5/Zn/PM**/Dox showed both a significant blue and red fluorescence. All the above results confirmed that all the obtained pillar [5]arene-based materials not only can be uptaken by HeLa cells efficiently, but also can be applied in living cell imaging.

### 2.5. In Vitro Cancer Therapy

The HeLa cells were also selected to investigate the cancer therapy effect of **TP5/Zn/PM**/Dox in vitro. After incubating with different groups, their viabilities were investigated via 3-(4,5-dimethylthiazol-2-yl)-2,5-diphenyltetrazolium bromide (MTT) assays [38,39,40]. As shown in Figure 5a, when the HeLa cells cultivated with **TP5/Zn**, **PM** and **TP5/Zn/PM** (concentration from 0–56 μg mL^−1^), the viabilities of the cells were all above 97%, indicating that our material itself has a good biocompatibility. For the cells in the **TP5/Zn/PM**/Dox and Dox groups, the cell viabilities all decreased as the concentration increased, and the cytotoxicity of **TP5/Zn/PM**/Dox was higher than that of Dox at the same concentration (Figure 5b). Nevertheless, the cell viability of **TP5/Zn/PM**/Dox was only 4% when the concentration increased to 60 µg/mL, exhibiting the largest cytotoxicity toward the HeLa cells.

At last, to check the cancer therapy effect of **TP5/Zn/PM**/Dox, live (green) and dead (red) cells were differentiated by calcein acetoxymethyl (calcein-AM) and propidium iodide (PI) staining. In the control and **TP5/Zn/PM** groups, the cells exhibited a green fluorescence, indicating that they are living well. On the contrary, when treated with free Dox, the cells exhibited an orange fluorescence. However, when treated with **TP5/Zn/PM**/Dox, almost all of the cells showed a bright red fluorescence, indicating that all of the cells died. These results clearly confirmed the satisfied therapeutic effect of **TP5/Zn/PM**/Dox.

## 3. Materials and Methods

All reagents were commercially available and used as supplied without further purification. Solvents were either employed as purchased or dried according to the procedures described in the literature. The ^1^H- or ^13^C-NMR spectra were recorded with a Bruker Avance DMX 400 spectrophotometer (Bruker, Bremen, Germany) with use of the deuterated solvent as the lock and the residual solvent or TMS as the internal reference. The solid-state nuclear magnetic resonance (NMR) spectra were recorded on a BRUKER 400WB AVANCE III spectrometer. A scanning electron microscopy (SEM) investigation was carried out on a JEOL 6390LV instrument (ZEISS, Oberkochen, Germany). The transmission electron microscopy (TEM) images were obtained using a Talos F200X instrument with an accelerating voltage of 80 kV (FEI, Hillsboro, OR, USA). UV-Vis spectroscopy was measured on a Shimadzu UV-2501 PC UV-Vis spectrometer (Shimadzu, Kyoto, Japan). The HeLa cells was purchased from Tongpai (Shanghai, China) Biotechnology Co., Ltd.

## 4. Conclusions

In brief, a terpyridine-modified pillar [5]arene (**TP5**) has been synthesized. Terpyridine was not only complexed with Zn ions but also endowed **TP5** with fluorescent properties. At the same time, the cavity of pillar [5]arene can complex with the guest molecule **PM** to form a stable supramolecular amphiphile. The resulting **TP5**/Zn/**PM** can further self-assemble into fluorescent particles. The obtained fluorescent particles can load and control the release of anti-cancer drugs effectively, which realized the precise release of drugs and living cell imaging. This work may provide a new way for scientists to construct nano-theranostics through dynamic host–guest interactions.

## Data Availability

No applicable.

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
