# Peer review of "Nano-Theranostics Constructed from Terpyridine-Modified Pillar [5]arene-Based Supramolecular Amphiphile and Its Application in Both Cell Imaging and Cancer Therapy"

_molecules, 2022, doi:10.3390/molecules27196428_

Round 1
Reviewer 1 Report
The manuscript molecules-1905300 "Nano-theranostics constructed from terpyridine modified pillar[5]arene based supra-molecular amphiphile and its application in both cell imaging and cancer therapy" describes preparation of some novel pillar[5]arene-transition metal complexes and micelles based on them, which can be loaded and release doxorubicin in a controlled manner.
This is a concise manuscript that looks to an extent raw. Improvements are required both in the experimental part, and a more thorough check for typos in the main text and supplementary material.
I have the following comments and suggestions for the authors:
1) Too much attention is paid to the analysis of Prof. Wang’s work in introduction. It is necessary to shorten this fragment and add an analysis of 2-3 more works on this topic. Authors are advised to revise introduction. Moreover there is a lot of self-citation for Yong Yao (25%). It must be corrected. Some new articles on the design of pillar[5]arenes should be added. For example, Chem. Commun. 2021, 57(87), 11443-11456; Chem. Commun. 2022, 58(20), 3255-3269; New J. Chem. 2022, 46(5), 2033-2037; Bioorgan. Med. Chem. 2022, 57, 116649.
2) Why did the authors use “supra-molecular”? Is it neologism or misprint?
3) In abstract the authors wrote that the vesicles diameter is about 200 nm but in the main partof manuscript the size is 140-150 nm.
4) Why did the authors choose the pillar complex with zinc among the metals studied in the manuscript? Fe and Mn are also play an indispensable role in many life processes.
5) There is mistake in the synthetic scheme of PM in supplementary material. The Scheme does not match the text. It must be corrected.
6) How did the authors can explain relatively low yields for all synthesized pillars (<45%)? Is it possible to increase the yields?
7) Is there single crystal structure for compounds TP5-1 and TP5-2? Does it agree with the NMR data for TP5-1 and TP5-2?
8) Why did the authors choose TP5-3 for investigation metal binding?
9) –line 84- “the fluorescence intensity decreased sharply after addition of Ru3+, Fe3+, Zn2+”. There are no significant changes for the complex with Ru in the spectrum according to the picture.
10) Did the authors define association constant for complexes? Association constants must be specified. Why the titration was performed only for zinc? The change with Zn is not the biggest in the spectrum.
11) The authors provide single crystal structure in figure 2d for the complex with palladium, but changes for palladium are minimal. What did the authors want to show by this picture?
12) In what solvent was the NMR spectrum of TP5-3/Zn?PM1 recorded?
13) The shifts of proton signals must be signed for each group. In supplementary material the authors must to sign each group in NMR spectrum of complex. What other methods can the authors confirm the formation of [2]pseudorotaxane?
14) The authors claimed that they obtain vesicles in 2.3. The figure 3 show micelles, since there is no difference between the center and the periphery according to the TEM. How was the formation of vesicles confirmed? Did the authors determine the concentration at which vesicle formation occurs?
15) The SEM cannot confirm the formation of vesicles cause the SEM shows only surface of particles. It must be corrected.
16) What is the confidence range for the diameter determined by the DLS? What is the polydispersity index for this systems?
17) Why the zeta-potential is negative according to the authors?
18) How was the drug loaded into particles?
19) The conclusion must be rewritten.
Author Response
Referee 1.
- Too much attention is paid to the analysis of Prof. Wang’s work in introduction. It is necessary to shorten this fragment and add an analysis of 2-3 more works on this topic. Authors are advised to revise introduction. Moreover there is a lot of self-citation for Yong Yao (25%). It must be corrected. Some new articles on the design of pillar[5]arenes should be added. For example, Chem. Commun. 2021, 57(87), 11443-11456; Chem. Commun. 2022, 58(20), 3255-3269; New J. Chem. 2022, 46(5), 2033-2037; Bioorgan. Med. Chem. 2022, 57, 116649.
Response: Thank you. We have added an analysis of 2-3 more works on this topic, and deleted some self-citation. Some new articles on the design of pillar[5]arenes also be added.
- Why did the authors use “supra-molecular”? Is it neologism or misprint?.
Response: Thank you. We have corrected “supra-molecular” to “supramolecular”
- In abstract the authors wrote that the vesicles diameter is about 200 nm but in the main part of manuscript the size is 140-150 nm.
Response: Thank you. We have made a mistake in the abstract. We have corrected it in our revised manuscript.
- Why did the authors choose the pillar complex with zinc among the metals studied in the manuscript? Fe and Mn are also play an indispensable role in many life processes.
Response: Thank you. Zinc plays an indispensable role in many life processes, it can promote the metabolism of various substances in the body, promote the proliferation of the lymphatic system, enhance the resistance to viruses and bacteria, etc. Our next work will focus on Fe and Mn complex.
- There is mistake in the synthetic scheme of PM in supplementary material. The Scheme does not match the text. It must be corrected
Response: Thank you. We have corrected it in our revised manuscript.
- How did the authors can explain relatively low yields for all synthesized pillars (<45%)? Is it possible to increase the yields?
Response: Due to the phenol is easily oxidized in air, so the yield is not high. We then used N2 to protect this reaction, and the yield can be improve to about 72%.
- Is there single crystal structure for compounds TP5-1 and TP5-2? Does it agree with the NMR data for TP5-1 and TP5-2?
Response: There is no single crystal structure for compounds TP5-1 and TP5-2, and we have deleted TP5-1 and TP5-2 in our revised manuscript.
- Why did the authors choose TP5-3 for investigation metal binding?
Response: TP5-1, TP5-2 and TP5-3 are structurally similar, so one of them was chosen to be combined with zinc for monitoring. We have deleted TP5-1 and TP5-2 in our revised manuscript.
- –line 84- “the fluorescence intensity decreased sharply after addition of Ru3+, Fe3+, Zn2+”. There are no significant changes for the complex with Ru in the spectrum according to the picture.
Response: Thank you. We have made a mistake here, and we have deleted the part about TP5-3 associated with different ions in our revised manuscript.
- Did the authors define association constant for complexes? Association constants must be specified. Why the titration was performed only for zinc? The change with Zn is not the biggest in the spectrum.
Response: Thank you. In our revised manuscript, we deleted the part of TP5 coordination with different metal ions due to we just investigated Zn afterward.
- The authors provide single crystal structure in figure 2d for the complex with palladium, but changes for palladium are minimal. What did the authors want to show by this picture?
Response: Thank you. We have reorganized the manuscript and deleted this single crystal structure.
- In what solvent was the NMR spectrum of TP5-3/Zn?PM1 recorded?
Response: The solvent is CDCl3.
- The shifts of proton signals must be signed for each group. In supplementary material the authors must to sign each group in NMR spectrum of complex. What other methods can the authors confirm the formation of [2]pseudorotaxane?
Response: The shifts of proton signals for each group was signed in Figure S5. Here we coold not obtained the crystal structure of the [2]pseudorotaxane, so we just used NMR to characterize the formation of [2]pseudorotaxane .
- The authors claimed that they obtain vesicles in 2.3. The figure 3 show micelles, since there is no difference between the center and the periphery according to the TEM. How was the formation of vesicles confirmed? Did the authors determine the concentration at which vesicle formation occurs?
Response: Thank you. It is most like particles from the TEM images, and we have corrected it. The CAC value is 2.32 ´ 10-6 mol/L from surface tension change.
- The SEM cannot confirm the formation of vesicles cause the SEM shows only surface of particles. It must be corrected.
Response: Thank you. We have corrected it.
- What is the confidence range for the diameter determined by the DLS? What is the polydispersity index for this system?
Response: Thank you. The confidence range for the diameter determined by the DLS was not recorded. The polydispersity index for this system is 0.38.
- Why the zeta-potential is negative according to the authors?
Response: Thank you. Zeta-potential is negative due to the poly-PEG chain
- How was the drug loaded into particles?
Response: Thank you. Firstly, 0.0066 g of TP5/Zn was dissolved with 1 mL DMSO to obtain the solution with a concentration of 1 × 10-3 mol/L. Then 50 µL solution was added into 5mL-volumetric flask and added water to 5mL to obtain the aqueous solution with concentration 10-4 mol/L. A proper amount of PM solution was added into the aqueous solution of TP5/Zn, and the ratio of TP5/Zn to PM was 1:1.2. The ultrasound was conducted at room temperature for 20 min, then the same amount of DOX solution was added, and the ultrasound was continued for 20 min. The mixed solution of TP5/Zn/PM/DOX was dialyzed in ultrapure water for several times, and the uncoated DOX and organic solvent DMSO were removed to prepare the aqueous solution of TP5/Zn/PM/DOX nanoparticles.
- The conclusion must be rewritten.
Response: Thank you. We rewrote the conclusion in our revised manuscript.
Reviewer 2 Report
In this paper, the researchers have designed and synthesized a series of ter- 12 pyridine-modified-pillar [5] arenes (TP5). By the coordination of terpyridine and metal ions, a series 13 of pillar [5] arene-transition metal complexes (TP5-MCs) were obtained. Then, supramolecular am- 14 phiphile can be constructed by using host-guest complexation between a polyethylene glycol con- 15 tained guest (PM) and TP5-3/Zn. Combined the fluorescence properties from terpyridine group and 16 its amphiphilicity, the obtained TP5-3/Zn/PM can further self-assembled into fluorescent vesicles 17 with diameters of about 200 nm in water. The obtained vesicles can effectively load and controlled 18 release of anti-cancer drugs and realized precise release of drugs and living cell imaging. This work is of great significance.
However, there are three points to be improved:
1.Please supplement 3 literatures in recent 3 years.
2. Please provide the source where you obtained HeLa cells.
3.Please describe the details of culturing HeLa cells in the main article.
Author Response
Referee 2.
- Please supplement 3 literatures in recent 3 years.
Response: Thank you. We have added some literatures in recent 3 years.
.
- Please provide the source where you obtained HeLa cells.
Response: HeLa cells was purchased from Tongpai (Shanghai) Biotechnology CO., LTD.
- Please describe the details of culturing HeLa cells in the main article.
Response: HeLa cells were incubated in Dulbecco’s modified Eagle’s medium (DMEM). The medium was supplemented with 10% fetal bovine serum and 1% Penicillin-Streptomycin. HeLa cells were seeded in 96-well plates (5 × 104 cell mL–1, 0.1 mL per well) for 24 h at 37oC in 5% CO2. Then the cells were incubated in TP5/Zn/PM, Dox and TP5/Zn/PM/Dox for 4 h, respectively. The medium was then removed, and the cells were washed 3 times with phosphate buffer. Finally, the cells were observed by fluorescence microscopy.
Round 2
Reviewer 1 Report
The manuscript of Chao-Guo Yan, Yong Yao and coauthors looks much better.
All neccesary changes have been made.
So I recommended these manuscript to publication in Molecules.